# Two-step self-assembly of a spider silk molecular clamp

Charlotte Rat[1], Julia C. Heiby[1], Jessica P. Bunz[1,2] & Hannes Neuweiler [1]

Web spiders synthesize silk fibers of unique strength and extensibility through the controlled self-assembly of protein building blocks, so-called spidroins. The spidroin C-terminal domain is highly conserved and connects two polypeptide chains through formation of an all-helical, intertwined dimer. Here we use contact-induced fluorescence self-quenching and resonance energy transfer in combination with far-UV circular dichroism spectroscopy as three orthogonal structural probes to dissect the mechanism of folding and dimerization of a spidroin C-terminal domain from the major ampullate gland of the nursery web spider *Euprosthenops australis*. We show that helices forming the dimer core assemble very rapidly and fold on association. Subsequently, peripheral helices fold and dock slowly onto the preformed core. Lability of outer helices facilitates formation of a highly expanded, partially folded dimer. The high end-to-end distance of chain termini in the partially folded dimer suggests an extensibility module that contributes to elasticity of spider silk.

---

[1] Department of Biotechnology & Biophysics, Julius-Maximilians-University Würzburg, Am Hubland, 97074 Würzburg, Germany. [2] Present address: Spiber Technologies AB, AlbaNova University Center, SE-10691 Stockholm, Sweden. Correspondence and requests for materials should be addressed to H.N. (email: hannes.neuweiler@uni-wuerzburg.de)

Web spiders use up to seven specialized glands to synthesize silk fibers of outstanding mechanical properties tailored for various tasks including prey capture, reproduction, and shelter[1]. Spider silk is characterized by a unique combination of strength and extensibility, which results in toughness that supersedes that of man-made threads[2–4]. At the beginning of the process the constituting protein building blocks, so-called spidroins, are stored in soluble form and at high concentration in the ampulla of the spinning gland, which is located in the spider's abdomen. On demand, silk is formed by the controlled self-association of spidroins within a tapering spinning duct. Chemical and mechanical stimuli within the duct induce phase and structural transitions of spidroins that transform into silk[1,3,4].

Spidroins from the major ampullate (Ma) gland form dragline silk, which is the toughest fiber and used by spiders as a lifeline or to build the web frame. Ma silk is a current focus of biomimetic material sciences[3,4]. The central segment of Ma spidroins consists of repetitive peptide motifs of simple amino acid composition involving alanine-, glycine-, and proline-rich stretches, which are unstructured under storage conditions and form mainly β-sheet structure in fibers[4]. The N- and C-terminal domains (NTD and CTD) flank the repetitive core domain and are highly conserved across glands and species. Conservation underscores their important functional roles in silk. A recent genomic study shows that sequences of CTDs are slightly more diverse compared to those of NTDs[5]. In particular, sequences from the pyriform and aggregate glands differ from those of other glands, which may reflect the fact that pyriform and aggregate silk has adhesive rather than fiber-forming function[5].

Both NTD and CTD are regularly folded five-helix bundles that provide solubility and connectivity of spidroins[6–8]. The NTD forms an antiparallel dimer, a process that is induced by a change of pH and salt composition along the spinning duct. A comprehensive body of biophysical work shows that pH-triggered dimerization of NTDs involves changes of protein surface electrostatics and conformation[3,6,8–14]. The CTD, on the other hand, forms a constitutive, parallel dimer already in the ampulla of the gland and connects two spidroin chains permanently. Although NTD and CTD share the same secondary structure they are very different on the level of tertiary and quaternary structure, highlighting their different functional roles. In contrast to the NTD, the fold of the CTD is intertwined: the C-terminal ends of each subunit undergo a domain swap and dock onto the core helix of the opposing subunit[7]. The topology of the fold resembles a molecular clamp. Parallel orientation of CTD subunits aligns the appended, repetitive segments, prestructuring and priming them for shear-induced phase and structural transitions within the duct[7,15]. Ma spidroin CTDs contain a single, conserved cysteine that forms a disulfide connecting two core helices of the dimer interface covalently. Structural studies show that the clamp-like fold is highly conserved across glands and species[16–18] despite the lack of a disulfide in CTDs of non-Ma spidroins[19,20]. Conservation of structure underscores a common functional role.

Here, we investigate folding/unfolding transitions associated with self-assembly of the CTD of spidroin 1 from the Ma gland (MaSp1) of the nursery web spider Euprosthenops australis. We use site-directed mutagenesis and chemical modification to establish extrinsic fluorescence reporter systems based on resonance energy transfer (FRET) and contact-induced self-quenching tailored to probe specific conformational coordinates. We apply steady-state spectroscopy and kinetic experiments to probe structure, dynamics and energetics of folding and dimerization yielding mechanistic insights. We identify a three-state mechanism of self-assembly that involves a partially folded, dimeric intermediate. In the first step, the unfolded chains of two monomers associate to form the helical core of the dimerization interface at remarkably fast rate constant. In the second step, the peripheral helices fold slowly onto the preformed core. Lability of peripheral helices facilitates formation of a highly expanded structure that may contribute to elasticity of spider silk.

## Results

**Identification of a folding intermediate.** We synthesized the MaSp1 CTD from *E. australis* through heterologous overexpression in *Escherichia coli* bacterial cells followed by chromatographic purification. We characterized the secondary structure of the domain and its folding/unfolding transitions using far-ultraviolet (UV) circular dichroism (CD) spectroscopy in combination with equilibrium chemical denaturation experiments. We used urea as denaturant in phosphate buffered solution at pH 7.0 and probed the helical content by measuring the CD signal at 222 nm where the ellipticity of α-helix is maximal. Under nonreducing conditions, where the intermolecular disulfide is retained, we observed a cooperative two-state transition between the natively folded dimer and a partially folded denatured state that retained ~50% helix (Fig. 1a). Under reducing conditions, where the disulfide bond is broken, we observed an additional unfolding transition at higher denaturant mid-point concentration ($[urea]_{50\%}$) (Fig. 1a). To assess reversibility of these transitions we performed renaturation experiments. To this end, CTD samples were initially fully denatured under reducing conditions in 10 M urea followed by renaturation in solutions of progressively reduced urea concentration. Renaturation data overlaid well with those of denaturation showing that the unfolding transitions were fully reversible (Fig. 1b).

We performed thermal denaturation experiments using far-UV CD spectroscopy, complementing chemical denaturation. In thermal denaturation experiments carried out under nonreducing conditions we observed unfolding of the native dimer in two discrete steps. Although the CTD containing a disulfide bond resisted full unfolding in chemical denaturation experiments, thermal denaturation lead to complete unfolding of residual structure at a mid-transition temperature of ~80 °C (Fig. 1c). But, in contrast to chemical denaturation, thermal denaturation was irreversible. We thus omitted analysis of thermal denaturation data using thermodynamic models that rely on reversibility.

To further characterize the nature of the observed intermediate we performed chemical denaturation experiments under reducing conditions and at varying protein concentration. We found that $[urea]_{50\%}$ of the second transition increased with increasing protein concentration while $[urea]_{50\%}$ of the first transition remained essentially invariant (Fig. 1d). Data recorded under reducing conditions fitted well to a thermodynamic model for a three-state folding equilibrium involving a dimeric intermediate[21] and the linear-free energy relationship[22,23] (Fig. 1d). Results were consistent with a model of the first transition reflecting a monomolecular folding event and the second transition reporting on a folding event that is associated with a bimolecular monomer/dimer equilibrium, which depends on protein concentration. Analysis revealed a populated intermediate that was a partially folded CTD dimer. The first transition was essentially invariant on protein concentration, which was unfolding of the native dimer ($N_2$) to the dimeric intermediate ($I_2$), and had an equilibrium *m*-value of $m_{N2–I2} = 1.3 \pm 0.2$ kcal mol$^{-1}$M$^{-1}$ and a $[urea]_{50\%}$ value of $2.6 \pm 0.1$ M (±s.d. of four measurements). The corresponding change of free energy was $\Delta G_{N2–I2} = 3.4 \pm 0.6$ kcal mol$^{-1}$. The second transition, which reflected cooperative unfolding and dissociation of the intermediate ($I_2$) forming denatured monomers (D), had a free energy of $\Delta G_{I2-D} = 14 \pm 1$ kcal mol$^{-1}$. The total free energy of two-step self-assembly of the

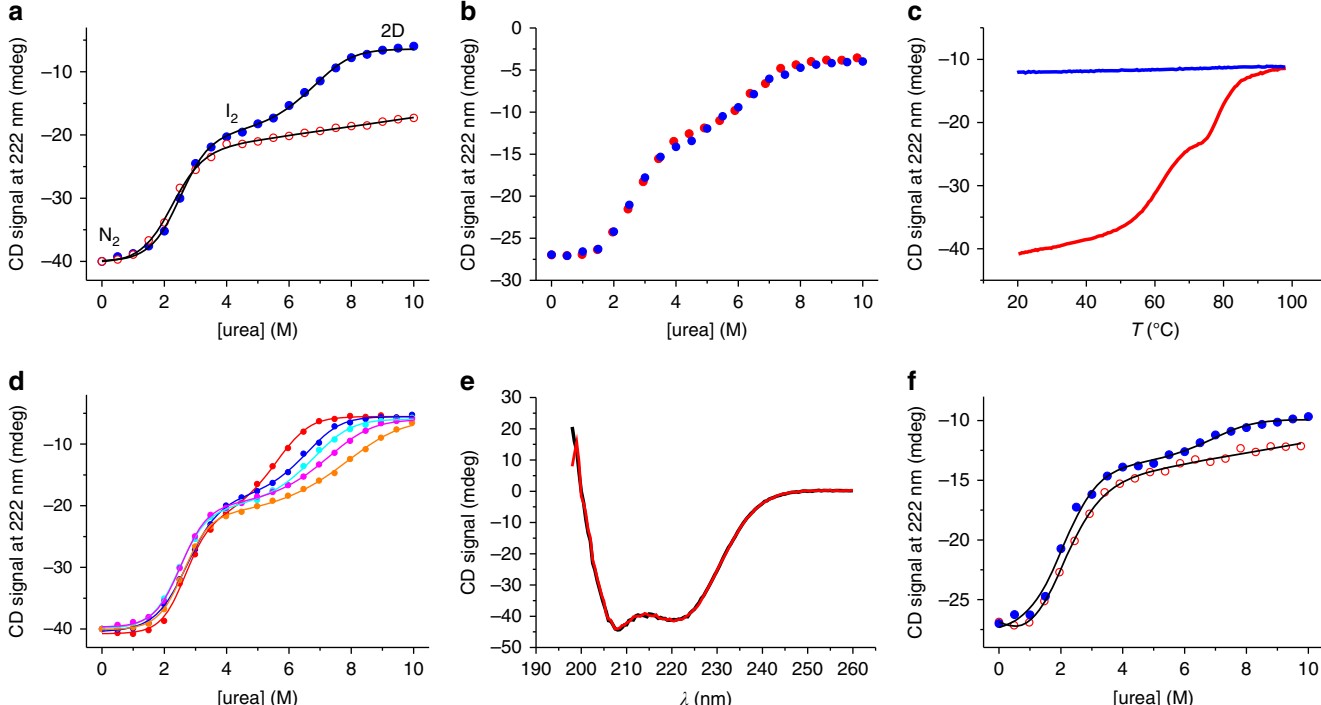

**Fig. 1** Equilibrium denaturation measured by far-UV CD spectroscopy. **a** Equilibrium chemical denaturation of 15 µM CTD measured under reducing (blue filled circles) and under nonreducing (red open circles) conditions at pH 7.0. Black lines are fits to the data using a thermodynamic model for a two-state transition (in case of nonreducing conditions) and for a three-state transition involving a dimeric intermediate (in case of reducing conditions). The nature of populated states is indicated. **b** Renaturation of 10 µM CTD, denatured in 10 M urea and refolded by dilution into pH 7.0 buffer (blue solid spheres). Corresponding denaturation data are shown for comparison (red solid spheres). **c** Thermal denaturation of 15 µM CTD measured under nonreducing conditions at pH 7.0 (red). A refolding trial measured by cooling the thermally denatured sample is shown in blue. **d** Chemical denaturation of CTD measured under reducing conditions and at varying protein concentration. Data recorded from 5 (red circles), 10 (blue circles), 15 (cyan circles), 20 (magenta circles), and 30 µM (orange circles) CTD are shown. Data are normalized to the native-state signal of 15 µM CTD for reasons of clarity. Lines are fits to the data using the thermodynamic model for a three-state transition involving a dimeric intermediate. **e** Far-UV CD spectra of CTD recorded under non-reducing conditions in pH 7.0 (black line) and in pH 5.7 (red line) buffered solutions. **f** Chemical denaturation of 10 µM CTD recorded at pH 5.7 under reducing conditions (blue closed circles) and under nonreducing conditions (red open circles). Black lines are fits to the data using thermodynamic models for a three-state and for a two-state transition, respectively

CTD ($\Delta G_{N2-D} = \Delta G_{N2-I2} + \Delta G_{I2-D}$) was thus $17 \pm 2$ kcal mol$^{-1}$. Thermodynamics of the first unfolding transition measured under reducing conditions matched that of the unfolding transition observed under nonreducing conditions (Fig. 1a). Two-state unfolding under nonreducing conditions had values of $m_{N2-I2}$, [urea]$_{50\%}$ and $\Delta G_{N2-I2}$ of $1.2 \pm 0.2$ kcal mol$^{-1}$ M$^{-1}$, $2.2 \pm 0.1$ M and $2.6 \pm 0.5$ kcal mol$^{-1}$, respectively (±s.e. from regression analysis). From the relative CD signal amplitude of the first and of the second unfolding transition we estimated the loss of helical structure associated with formation of the intermediate to $58 \pm 3\%$.

The lowest pH measured in the very distal part of a spinning duct is 5.7[17]. In order to assess the structural integrity of the CTD at this pH we compared far-UV CD spectra recorded at pH 7.0 and at pH 5.7. The spectra overlaid very well showing that the domain remained fully folded at pH 5.7 (Fig. 1e). We carried out chemical denaturation experiments at pH 5.7 under reducing and under non-reducing conditions (Fig. 1f). The first transition at pH 5.7 had a lower [urea]$_{50\%}$-value and a lower $m$-value compared to at pH 7.0 ([urea]$_{50\%} = 1.8 \pm 0.2$ M and $m_{N2-I2} = 0.9 \pm 0.1$ kcal mol$^{-1}$ M$^{-1}$ at pH 5.7). The native dimer was thus destabilized at low pH ($\Delta G_{N2-I2} = 1.6 \pm 0.3$ kcal mol$^{-1}$ at pH 5.7). The observation may be explained by weakening of the salt bridge formed by residues Arg42 and Glu91, which stabilizes N-terminal helix 2 of the fold, at low pH through protonation. At pH 5.7 and

under reducing conditions there was significant residual ellipticity at urea concentrations >8 M (Fig. 1f). The chemically denatured state thus appeared more structured at pH 5.7 compared to at pH 7.0.

**The partially folded intermediate is highly expanded.** In order to probe intramolecular distance changes associated with folding/unfolding we applied FRET spectroscopy. Nonradiative FRET between a donor and an acceptor fluorophore has a characteristic $1/r^6$ distance-dependence and serves as a molecular ruler widely applied to biology[24]. The Förster distance of the donor/acceptor FRET pair Alexa-Fluor-488/Alexa-Fluor-594, i.e., the inter-molecular fluorophore distance at which the FRET efficiency drops to 50%, is $R_0 = 6$ nm[25]. We generated a homology model of the CTD from *E. australis* using the protein structure homology-modeling server SWISS-MODEL[26] and the available structural data of a spidroin CTD from *Araneus diadematus* (PDB id 2KHM)[7] as a template (Fig. 2a). We engineered a single cysteine (Cys) to site-specifically introduce donor/acceptor fluorophores through chemical thiol modification. To this end, we first removed the native Cys at sequence position 82 using site-directed mutagenesis (mutant C82A). We then introduced, on the background of mutant C82A, a single Cys at the N-terminal end of helix 1 for site-specific fluorescence modification (mutant C82A-S14C). The intermolecular distance of the side chain of S14

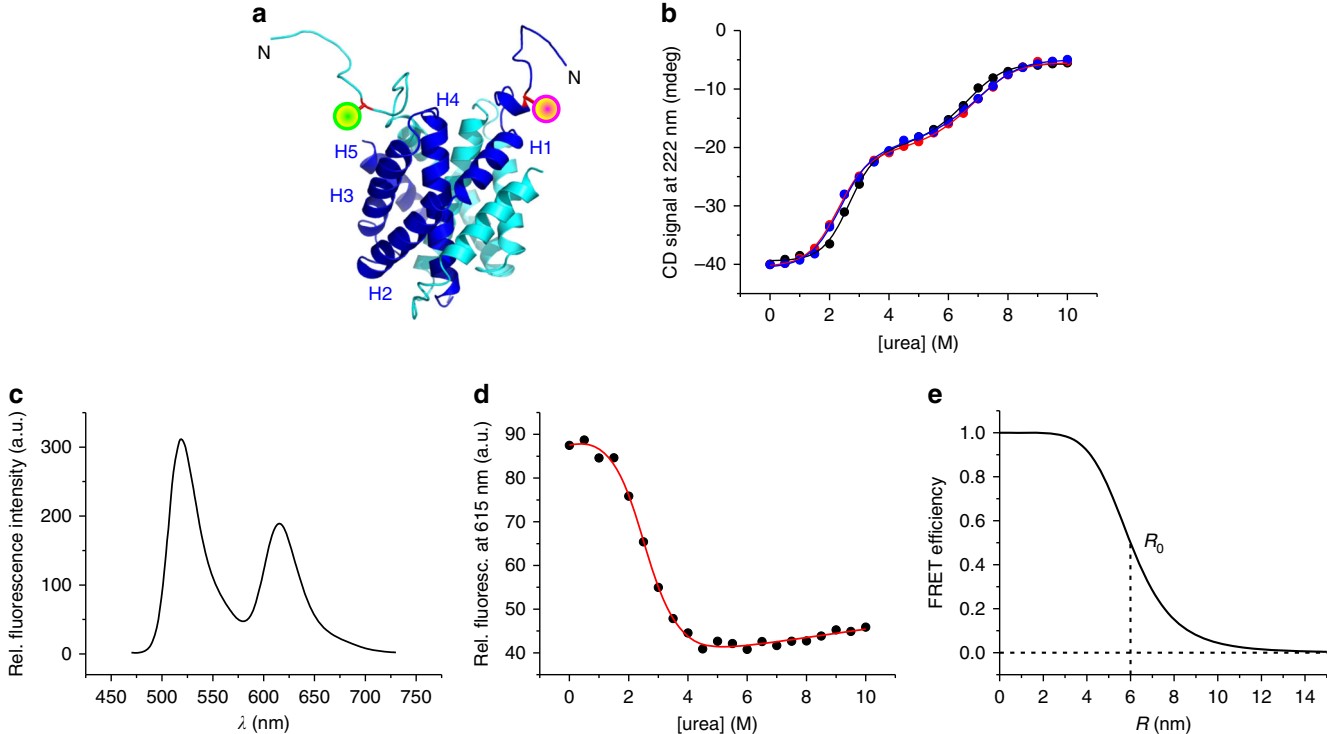

**Fig. 2** Unfolding of N-terminal helices probed by FRET. **a** Homology model of the MaSp1 CTD from *E. australis* (template: pdb id 2KHM) shown in cartoon representation. Subunits of the dimer are depicted blue and cyan. Helices 1–5 of one subunit are indicated. Mutant side chain S14C at the N-terminal end of helices, which was site-specifically modified with Alexa-Fluor-488 (green sphere) and Alexa-Fluor-594 (magenta sphere), is highlighted. **b** Chemical equilibrium denaturation of mutant C82A (red spheres) and mutant C82A–S14C (blue spheres) measured in pH 7.0 buffered solutions under nonreducing conditions. Denaturation data of wild-type CTD measured in pH 7.0 buffered solution under reducing conditions are shown for comparison (black spheres). Solid lines are fits to the data using a thermodynamic model for a three-state equilibrium involving a dimeric intermediate. **c** FRET spectrum of Alexa-Fluor-488/Alexa-Fluor-594-modified CTD recorded in pH 7.0 buffered solution. Fluorescence was excited at 460 nm. **d** Chemical equilibrium denaturation of Alexa-Fluor-488/Alexa-Fluor-594-modified CTD probed by measuring Alexa-Fluor-594 fluorescence emission intensities (black circles). The red line is a fit to the data using the thermodynamic model for a two-state equilibrium. **e** Theoretical FRET efficiency curve of the fluorophore pair Alexa-Fluor-488/Alexa-Fluor-594 plotted as function of intermolecular distance $R$. The Förster distance $R_0$ is indicated

in the CTD dimer is 3 nm (Fig. 2a), i.e., well below $R_0$ of Alexa-Fluor-488/Alexa-Fluor-594. Far-UV CD spectroscopy in combination with chemical denaturation experiments showed that mutations C82A and S14C did not cause structural or energetic perturbations: ellipticities and the two-step unfolding profile of the native CTD recorded under reducing conditions was quantitatively retained in mutants C82A and C82A–S14C (Fig. 2b). We modified mutant C82A–S14C with thiol-reactive derivatives of Alexa-Fluor-488 and Alexa-Fluor-594 and formed heterodimers consisting of Alexa-Fluor-488/Alexa-Fluor-594-modified CTD subunits. Photoexcitation of Alexa-Fluor-488 resulted in FRET to Alexa-Fluor-594, which was evident from steady-state fluorescence emission spectra where the donor Alexa-Fluor-488 was selectively excited and substantial fluorescence emission of the acceptor Alexa-Fluor-594 was detected (Fig. 2c). We measured acceptor fluorescence emission intensities of Alexa-Fluor-488/Alexa-Fluor-594-modified CTD samples at increasing concentrations of denaturant. The resulting chemical denaturation curve showed a single transition that fitted well to a thermodynamic two-state model (Fig. 2d). The obtained equilibrium $m$-value ($m = 1.0 \pm 0.1$ kcal mol$^{-1}$M$^{-1}$) and denaturant mid-point concentration ([urea]$_{50\%} = 2.5 \pm 0.1$ M) matched the values of the first unfolding transition of the nonmodified protein measured using far-UV CD spectroscopy ($m = 1.0 \pm 0.1$ kcal mol$^{-1}$ M$^{-1}$, [urea]$_{50\%} = 2.5 \pm 0.1$ M, Fig. 2b). The good agreement of thermodynamic quantities measured from modified and nonmodified

material showed that fluorescence modification did not perturb folding. We could thus assign with confidence the first unfolding transition of the CTD dimer observed by CD spectroscopy to unfolding of the N-terminal helices probed by FRET. Observation of a two-state transition by FRET spectroscopy also showed that FRET between Alexa-Fluor-488 and Alexa-Fluor-594 solely reported on formation of the partially folded dimer ($I_2$) and not on dissociation of $I_2$ forming unfolded monomers (D). Apparently, Alexa-Fluor-488 and Alexa-Fluor-594 located on N-terminal helices of $I_2$ were already too far apart to cause residual FRET signal that would be lost upon dissociation of $I_2$ forming D. The theoretical FRET efficiency $E$ of Alexa-Fluor-488/Alexa-Fluor-594 calculated as a function of intermolecular distance $R$, $E = 1/(1 + (R/R_0))^6$ (ref. [24]), predicted that FRET was fully lost at intermolecular distances >10–12 nm (Fig. 2e). We concluded that the N-terminal ends of unfolded helices in $I_2$ were separated by at least ~10 nm.

**N-terminal helices fold and dock slowly onto the dimer core.** Equilibrium far-UV CD spectroscopy captured events both of folding and dimerization (Fig. 1). We aimed at measuring the time course of both processes using rapid-mixing experiments combining far-UV CD spectroscopy with the stopped-flow technique. In initial experiments we refolded monomeric, chemically denatured CTD using the stopped-flow machine. We observed transients that showed a rapid burst kinetic phase,

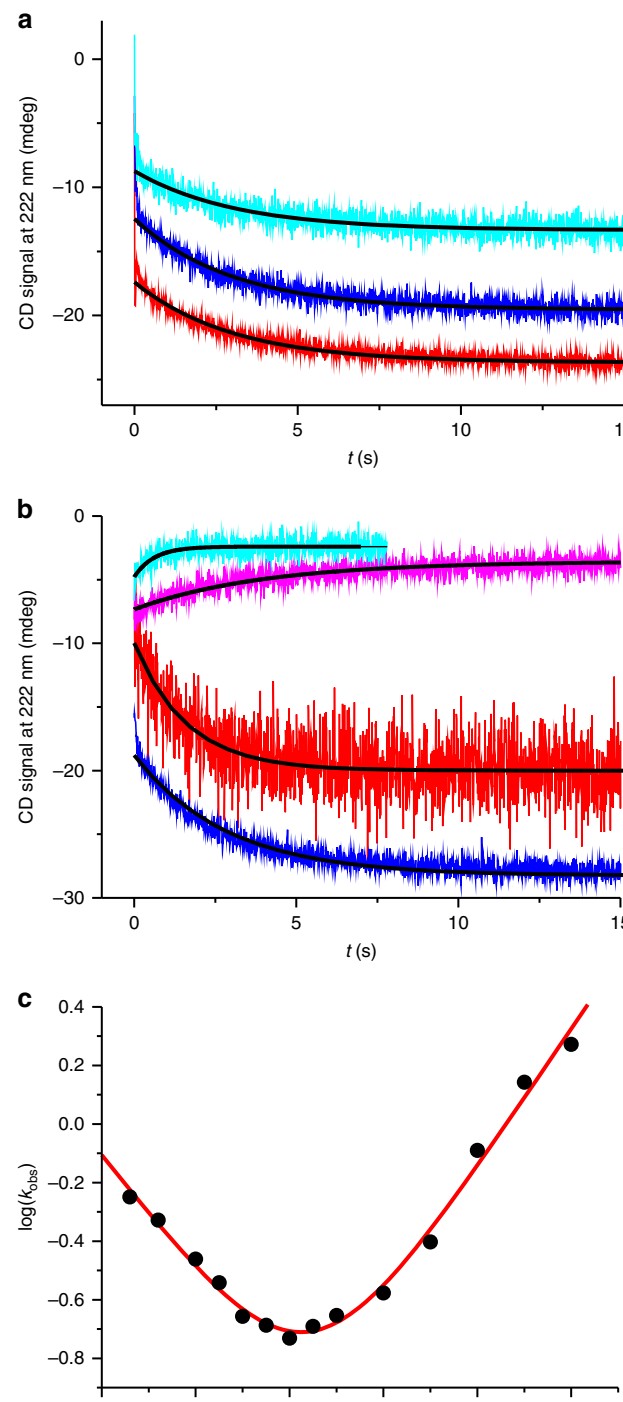

**Fig. 3** Folding kinetics of N-terminal helices. **a** Kinetic transients of refolding measured using far-UV CD stopped-flow spectroscopy. Chemically reduced and denatured wild-type CTD samples at 24 (cyan), 38 (blue), and 100 μM (red) protein concentration, refolded by rapid mixing into pH 7.0 buffered solution. Transients are offset along the y-axis for reasons of clarity. Black lines are fits to the data using a mono-exponential decay function. **b** Representative kinetic transients of folding (red: in 0.3 M urea; blue: in 1.0 M urea) and unfolding (magenta: in 2.5 M urea; cyan: in 4.5 M urea) of nonreduced wild-type CTD measured using far-UV CD stopped-flow spectroscopy. Transients are offset along the y-axis for reasons of clarity. Black lines are fits to the data using a mono-exponential decay function. **c** Observed rate constants plotted versus denaturant concentration (chevron plot). The red line is a fit to the data using the kinetic model for a barrier-limited two-state transition

which was followed by a slow, single-exponential relaxation (Fig. 3a). The burst phase occurred within ~10 ms, i.e., beyond the time resolution of our setup. The slow relaxation decayed on the seconds time scale. To get insight into the molecular nature of the slow phase we measured kinetics at various protein concentrations. Observed time constants obtained from exponential fits to the data recorded at 24, 38, and 100 μM CTD were 3.1 ± 0.1, 3.0 ± 0.1, and 3.0 ± 0.1 s, respectively, i.e., within error (Fig. 3a). The slow phase thus reported on a mono-molecular folding event that was independent on protein concentration. We speculated that the unresolved rapid burst phase reported on association of unfolded monomers forming the dimeric intermediate. Bimolecular dimerization appeared to be too fast to be measured at the high-protein concentrations required for sufficient signal-to-noise in far-UV CD spectroscopy.

In order to assign the slow exponential phase with confidence and to accurately determine rate constants of folding and unfolding, we measured kinetic transients at various concentrations of urea and under nonreducing conditions. We plotted the observed rate constants versus urea concentration and performed chevron analysis[27] (Fig. 3b, c). Chevron analysis revealed extrapolated, microscopic rate constants of folding and unfolding, which were $k_f = 0.78 \pm 0.07$ s$^{-1}$ and $k_u = 9 \pm 2 \times 10^{-3}$ s$^{-1}$. The free energy of folding calculated from kinetics was $\Delta G = -RT\ln(k_f/k_u) = 2.7 \pm 0.2$ kcal mol$^{-1}$. This value that was in very good agreement with the one determined by equilibrium chemical denaturation ($\Delta G_{N2-I2} = 2.6 \pm 0.5$ kcal mol$^{-1}$). The sum of kinetic m-values of folding and unfolding obtained from chevron analysis ($m_f = 0.56 \pm 0.06$ kcal mol$^-$ M$^{-1}$ and $m_u = 0.65 \pm 0.04$ kcal mol$^{-1}$M$^{-1}$) was in very good agreement with the equilibrium m-value ($m_{N2-I2} = 1.2 \pm 0.2$ kcal mol$^{-1}$ M$^{-1}$). The good agreement of quantities obtained from kinetic and equilibrium experiments lead us to assign with confidence the slow kinetic phase to the reversible, barrier-limited two-state folding transition between $I_2$ and $N_2$.

**The dimer scaffold assembles rapidly from unfolded monomers.** In order to resolve the fast kinetics of dimerization we engineered an extrinsic fluorescence reporter based on contact-induced self-quenching, which facilitated stopped-flow fluorescence spectroscopy at low protein concentration. The intermolecular disulfide formed by cysteine C82 is located at the N-terminal end of helix 4, which represents a solvent-exposed part of the dimerization interface (Fig. 4a). Modifying C82 did not perturb structure or energetics of folding, which was evident from far-UV CD experiments described above (Fig. 2b). We modified the chemically reduced side chain of C82 site-specifically using the thiol-reactive fluorescence label AttoOxa11. AttoOxa11 is an oxazine derivative that exhibits strong fluorescence quenching upon formation of H-type dimers in aqueous solution[12]. In the modified CTD dimer (CTD-AttoOxa11) C82 brings the AttoOxa11 label of each subunit in immediate vicinity. This was evident from steady-state absorption and fluorescence emission spectroscopy. The absorption spectrum of the monomer (chemically denatured CTD-AttoOxa11) showed the characteristic absorption band of the dye in the red spectral range with a peak at 660 nm (Fig. 4b). By contrast, the absorption spectrum of the dimeric CTD (CTD-AttoOxa11 in pH 7 buffered solution without denaturant) showed characteristics of an H-type fluorophore dimer with two major absorption peaks[12,28]. Dimer formation was accompanied by strong quenching of AttoOxa11 fluorescence (Fig. 4b). AttoOxa11 at position C82 thus sensitively and site-specifically reported on formation of the dimer scaffold formed by helices 4, i.e., formation of the dimeric intermediate $I_2$.

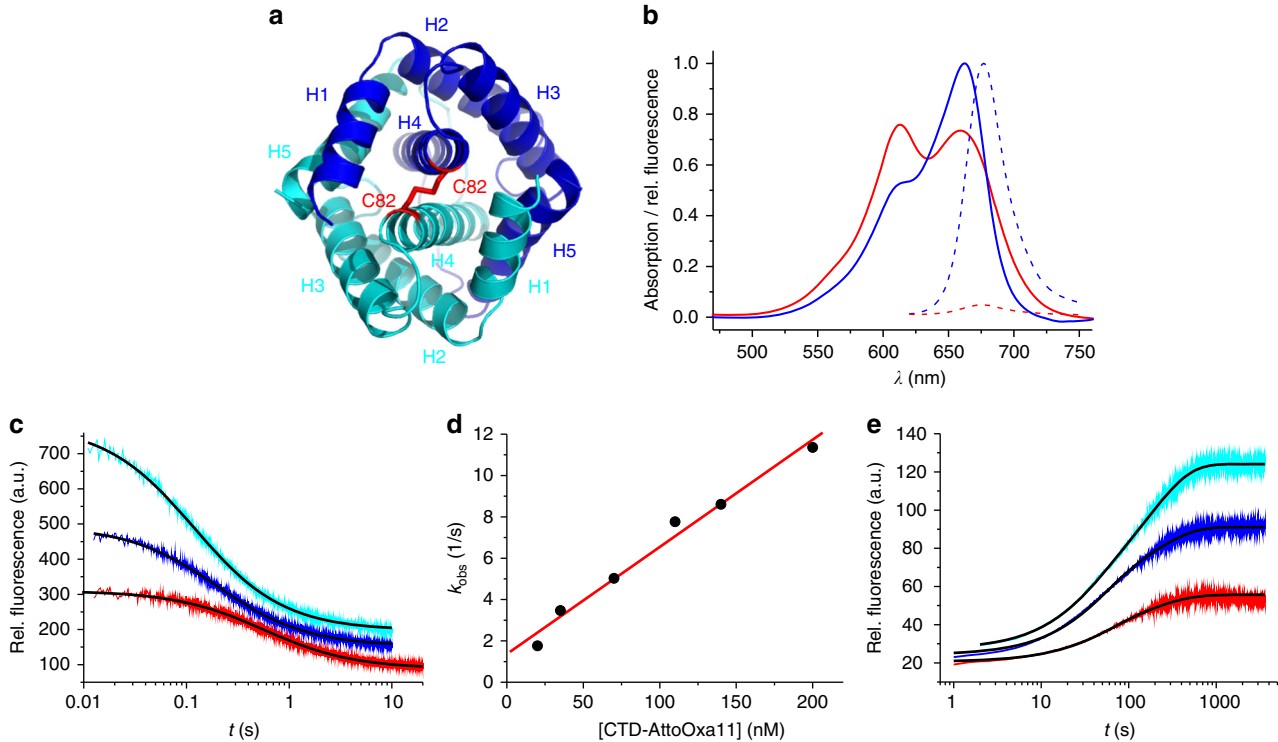

**Fig. 4** Kinetics of dimer formation and dissociation. **a** Top view on the homology model of the CTD shown in cartoon representation. The two subunits constituting the dimer are shown in blue and cyan. Helices 1–5 are indicated. The intermolecular disulfide formed by side chain C82, which was used for fluorescence modification, is highlighted in red stick representation. Unfolded N-terminal tails are omitted for reasons of clarity. **b** Steady-state absorption (solid lines) and fluorescence emission (broken lines) spectra of native dimeric (red) and chemically denatured monomeric (blue) CTD-AttoOxa11. **c** Kinetic transients of CTD dimerization measured by refolding chemically denatured CTD-AttoOxa11 using stopped-flow fluorescence spectroscopy. Representative transients of 20 (red), 70 (blue), and 140 nM (cyan) CTD-AttoOxa11 are shown. Transients are offset along the y-axis for reasons of clarity. Black lines are fits to a kinetic model of dimerization. **d** Observed rate constants of dimerization plotted versus concentration of CTD-AttoOxa11. The red line is a linear fit to the data. **e** Kinetic transients of CTD dissociation measured by chasing 60 (red), 120 (blue), and 180 nM (cyan) CTD-AttoOxa11 with 20 µM chemically reduced wild-type CTD. Transients are offset along the y-axis for reasons of clarity. Black lines are fits to the data using a bi-exponential function

We used stopped-flow fluorescence spectroscopy to measure kinetics of dimerization. Monomeric, chemically denatured CTD-AttoOxa11 was rapidly mixed into pH 7.0 buffer without denaturant using the stopped-flow machine. Kinetics was measured by recording the time course of AttoOxa11 fluorescence emission from sub-µM CTD-AttoOxa11 samples. The observed fluorescence transients were of large amplitude and fitted well to a kinetic model of dimerization (Fig. 4c). The observed rate constant increased linearly with protein concentration, as expected for a bimolecular event (Fig. 4d). From the linear fit to the data we determined the microscopic rate constant of dimerization to $k_{ass} = 5.2 \pm 0.4 \times 10^7 \, M^{-1} \, s^{-1}$.

To measure the rate constant of dissociation we performed chasing experiments. One fluorescently modified subunit in dimeric CTD-AttoOxa11 was chased off by rapidly mixing in excess of chemically reduced, nonmodified wild-type CTD. Since folding and dimerization of CTDs containing no disulfide was reversible and in dynamic equilibrium each dissociation event of AttoOxa11-modified CTD lead to rapid re-association with nonmodified CTD present at excess concentration. Dissociation of CTD-AttoOxa11 lead to dissociation of fluorophore dimers, which was signaled by an increase of AttoOxa11 fluorescence emission intensity. The time course of fluorescence intensity thus reflected the time course of CTD dissociation. The observed transients were of large amplitude and on a minutes time scale (Fig. 4e). Since dimer dissociation is a first-order reaction, the

underlying kinetics should follow a mono-exponential function[29]. Observed transients, however, required a sum of two exponentials to fit them accurately (Fig. 4e). The second exponential may be explained by molecular heterogeneity caused by, e.g., oligomerization, which was also observed for a spidroin NTD[12], or could reflect a two-step dissociation process. Observed rate constants, however, were essentially independent on protein concentration suggesting that formation of higher order oligomers or aggregates were not origin of the second exponential. Specifically, observed time constants (relative amplitudes) of the first exponential of 60, 120, and 180 nM CTD-AttoOxa11 was 51 (0.51), 37 (0.48), and 34 s (0.32), respectively. Observed time constants (relative amplitudes) of the second exponential of 60, 120, and 180 nM CTD-AttoOxa11 was 208 (0.49), 186 (0.52), and 194 s (0.68), respectively. We estimated the microscopic time constant of CTD dissociation by calculating the mean of the observed time constants weighted by their relative amplitudes to $\tau_{diss} = 128 \pm 14$ s. The mean rate constant of CTD dissociation was thus $k_{diss} = 8 \pm 1 \times 10^{-3} \, s^{-1}$.

From the microscopic rate constants $k_{ass}$ and $k_{diss}$ we calculated the free energy of dimerization ($\Delta G = -RT\ln(k_{ass}/k_{diss}) = 13.4 \pm 0.1 \, kcal \, mol^{-1}$). This value was within error of the free energy of dimerization obtained from equilibrium chemical denaturation of wild-type CTD measured using far-UV CD spectroscopy ($\Delta G_{I2\text{-}D} = 14 \pm 1 \, kcal \, mol^{-1}$). The good agreement of values obtained from modified and nonmodified protein material validated our

analysis and showed that fluorescence modification did not perturb dimerization.

## Discussion

Spidroins undergo phase and structural transitions within a spinning duct, which are induced by mechanical and chemical stimuli. The process involves a change of pH from 7.0 to 5.7 and has different effects on spidroin terminal domains[17]. In general, pH values above or below a protein's isoelectric point (pI) induce repulsive electrostatic forces between correspondent surface charges, which counteract protein–protein association. Most proteins have a pI below 7. As the solution pH approaches the pI, proteins tend to associate because net surface charges are neutralized and repulsive forces vanish. Web spiders seem to make use of this phenomenon to assemble spidroins along a pH gradient within their duct. At very low pH, however, many proteins tend to undergo acid-induced denaturation, which can lead to aggregation[30]. It may therefore not surprise that CTDs are destabilized at pH values below 7 (refs. [15,17] and [31,]). The CTD from the minor ampullate gland of *Araneus ventricosus* is reported to even form aggregates and β-sheet fibrils at low-pH values between 5.5 and 5.0[17]. MaSp1 CTDs of *Latrodectus hesperus* and *Nephila clavipes* form unstructured aggregates at pH 5 and below[15,31]. But in vitro aggregation of CTDs commonly takes hours or even longer and, in some cases, is induced only at highly acidic nonphysiological conditions. Web spiders synthesize silk fibers within a fraction of a second[3]. The relevance of observed CTD aggregation for silk formation by the animal is therefore unclear[15]. However, unfolding of the CTD during silking may induce transitions of functional relevance like the formation of β-sheet secondary structure or phase transition. In agreement with previous studies[15,17] we found that the MaSp1 CTD from *E. australis* is destabilized at low pH, but remained fully folded on the physiological time scale (Fig. 1e, f).

Our results showed that self-assembly of the CTD follows a three-state mechanism. This was evident from equilibrium and kinetic experiments where we observed two discrete structural transitions and determined the underlying rate constants and energetics. We deduced a mechanism of formation of the

molecular-clamp topology of the native CTD dimer from initially unfolded monomers (Fig. 5a). In the first step, the unfolded monomers associate to form the core helices of the dimer interface, i.e., the central scaffold built by helices 4. The rate constant of this process was remarkably fast ($k_{ass} = 5.2 \pm 0.4 \times 10^7 \, M^{-1} \, s^{-1}$). By comparison, the basal rate constant of protein–protein association is $\sim 1 \times 10^5 \, M^{-1} \, s^{-1}$ (refs. [32–34]), i.e., ~500 times slower than the value observed here. Theory assumes that two structures containing complementary association interfaces must orient in solution before they can form a productive complex in a lock-and-key fashion[32,33]. The entropic constraint of molecular orientation slows association to the basal value[32]. Intriguingly, the CTD dimer interface folds upon association. This was evident from the loss of helical secondary structure observed in the second transition of chemical denaturation data (Fig. 1d). Since folding and association are cooperative events, the entropic constraint of molecular orientation may drop out and explain fast dimerization. Alternatively, steering forces caused by long-range electrostatics may accelerate association beyond the basal value[33,34]. Ultrafast association is also observed for the MaSp1 NTD, which by contrast to the CTD proceeds from folded monomers[12,35].

The second step was folding and docking of the peripheral N-terminal helices onto the dimer scaffold. This process was on the surprisingly slow seconds time scale ($k_f = 0.78 \pm 0.07 \, s^{-1}$). Isolated helices are known to typically fold within a microsecond[36] and their docking can occur within a millisecond[37]. How can the dramatic slowing of helix folding and docking be rationalized? In the native CTD helix 5 undergoes a domain swap and threads through the chain of the opposing subunit where it clamps onto helix 4. The structure formed by helices 4 and 5 is the docking interface for N-terminal helices 1–3. The intertwined fold resembles a knotted protein. In knotted proteins, the knotting event limits the time constant of folding to the slow seconds-to-minutes scale[38–40]. Two-step folding of the CTD observed here resembles that of a dimeric domain from *Helicobacter pylori*, which also contains a molecular clamp and was engineered to knot[41]. Folding and docking of helices 1–3 in the CTD are thus likely to be rate-limited by domain swap, which involves threading and clamping of C-terminal helices 4 and 5. The rate

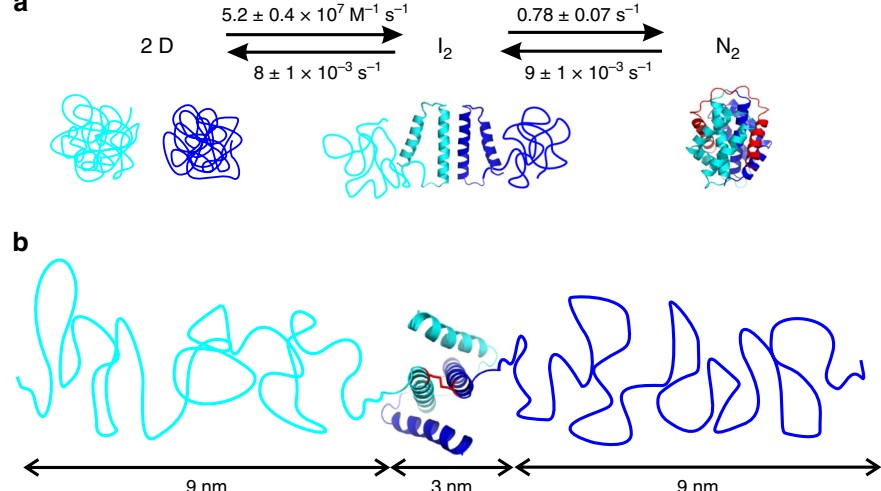

**Fig. 5** Reaction scheme of CTD self-assembly and structural model of the intermediate. **a** Reaction scheme, rate constants and structural model of two-step folding and dimerization of the CTD. The two subunits constituting the dimer are colored blue and cyan. Unstructured coils and helices are depicted in cartoon representation. C-terminal helix 5 in N$_2$, which swaps domain and threads through the chain of the opposing subunit, is highlighted red. **b** Structural model of the dimeric intermediate I$_2$. Chain dimensions of unfolded helices 1–3 inferred from FRET spectroscopy and polymer theory, and of the dimer core formed by helices 4 and 5, are indicated. The disulfide that covalently connects the N-terminal ends of helices 4 is highlighted in red stick representation

constant of CTD dissociation, i.e., of transition $I_2$ to $D$ ($k_{diss} = 8 \pm 1 \times 10^{-3}\,s^{-1}$), matched the rate constant of unfolding of N-terminal helices 1–3, i.e., of transition $N_2$ to $I_2$ ($k_u = 9 \pm 2 \times 10^{-3}\,s^{-1}$). This suggests that undocking/unfolding of peripheral helices limits the rate constant of dimer dissociation. This is reasonable because the peripheral helices have to first dismantle from the core before the chains can untangle to form unfolded monomers.

Why has the three-state mechanism of CTD self-assembly gone unnoticed in previous biophysical studies? Previously, unfolding of CTDs was described by a conventional two-state model[7,15,17,18,31]. But deviations from two-state behavior are reported: Andersson et al.[17] find that unfolding of a CTD from the minor ampullate gland of *A. ventricosus* deviates from two-state. Hagn et al.[7] report indications for early unfolding of helix 1 of a CTD from *A. diadematus* from NMR spectroscopy. Multistate mechanisms are difficult to detect if energetics of transitions are not well separated. In such cases, three-state mechanisms may appear two-state. But a third state reflecting a bimolecular event can be detected from protein concentration-dependent experiments[21,41,42]. We simulated theoretical denaturation curves using the three-state model applied to our experimental data. In this model we artificially reduced the free energy of dimerization ($\Delta G_{I2-D}$). As $\Delta G_{I2-D}$ drops from the here experimentally determined 14 kcal mol$^{-1}$ over 12.5 to 11.5 kcal mol$^{-1}$, the two discrete unfolding transitions of 10 µM CTD progressively overlay and appear two-state (Fig. 6).

Elasticity of spider silk is attributed to repetitive, glycine/proline-rich sequence areas that form helical structures in silk[43,44]. Infrared spectroscopy finds evidence for helix-to-coil transitions in spider silk upon expansion induced by mechanical strain[45]. But the CTD also plays a role: deletion of CTDs in engineered spidroins impairs extensibility of synthetic fibers[46]. The underlying molecular mechanisms remained elusive. Here we found that peripheral helices of the CTD are very labile ($\Delta G_{N2-I2} = 2.6 \pm 0.5$ kcal mol$^{-1}$) compared to the dimer core ($\Delta G_{I2-D} = 14 \pm 1$ kcal mol$^{-1}$). Imposing stress on a spider silk fiber will give rise to a stretching force on the N-terminal ends

of a CTD dimer where the central, repetitive spidroin domains append. This stretching force may induce early unfolding of labile CTD helices 1–3. We estimated expansion of the CTD upon unfolding of its N-terminal helices. The loss of helical content associated with the transition $N_2$ to $I_2$ determined by CD spectroscopy was $58 \pm 3\%$. This value was in good agreement with the fractional secondary structure of 54% formed by helices 1–3, as seen in the structure[7]. The Tanford $\beta$ value[47] of the folding of $I_2$ to $N_2$, revealed by chevron analysis, was low ($\beta_T = m_f/(m_f + m_u) = 0.46$). This showed that the transition state of folding of helices 1–3 was expanded and denatured-like, supporting an expanded N-terminus in $I_2$. FRET spectroscopy showed that the N-termini in $I_2$ exhibited an end-to-end distance of >10 nm (Fig. 2). The average end-to-end distance $\langle r^2 \rangle$ of an unfolded polypeptide chain can be estimated using polymer theory[48]. Helices 1–3 comprise a 64-residue polypeptide segment. An unfolded 64-residue segment has an end-to-end distance of ~9 nm ($\langle r^2 \rangle = C_\infty n l^2$)[48]. Here, $C_\infty$ is the characteristic ratio, which reflects chain stiffness and takes a value of ~9 for long, generic polypeptides[49], $n$ is the number of peptide bonds, and $l$ is the distance between two neighboring backbone amide bonds in a polypeptide ($l = 0.38$ nm). From polymer theory we can thus estimate that the two unfolded chain segments in $I_2$ form an extended structure of ~21 nm end-to-end distance (Fig. 5b). This corresponds to a ~sevenfold expansion of the CTD in state $I_2$ compared to in state $N_2$. A linear and fully stretched 64-residue polypeptide, which may be formed at high tensile stress just before the fiber breaks, has a length of $nl = 24$ nm. This corresponds to a 17-fold expansion of maximally stretched $I_2$ compared to $N_2$. In Ma silk, further expansion is blocked by the covalent disulfide between helices 4 (Fig. 5b).

In conclusion, we identified a two-step mechanism of self-assembly of the CTD. Fast folding and dimerization of central helix 4 is followed by slow domain swap and threading of the C-terminal helix forming the scaffold for docking of the N-terminal helices. The CTD self-assembles in the ampulla of the gland where the residence time of spidroins is sufficiently long for slow folding. On the other hand, slow kinetics of helix docking gives rise to their lability, which could facilitate partial unfolding in solid fibers induced by mechanical stress. The CTD may thus act as extensibility module that, in addition to helical central spidroin segments, contributes to elasticity of spider silk.

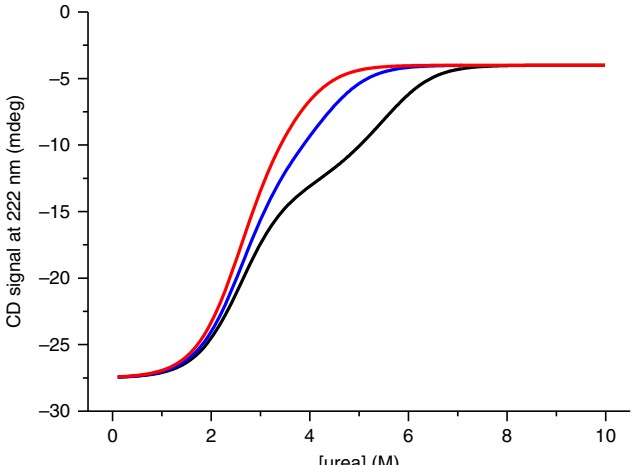

**Fig. 6** Theoretical denaturation curves of the CTD of varying free energy of dimerization. Simulated denaturation curves of 10 µM CTD using the model for a three-state equilibrium involving a dimeric intermediate. The curves contain the thermodynamic parameters derived from the fit to experimental data of wild-type CTD (free energy of dimerization: $\Delta G_{I2-D} = 14$ kcal mol$^{-1}$, black line). Simulated denaturation curves using the same model but lowering $\Delta G_{I2-D}$ to 12.4 and to 11.5 kcal mol$^{-1}$ are shown as blue and red line, respectively

## Methods

**Protein expression, mutagenesis, and modification**. The synthetic gene of the CTD of MaSp1 from the spider species *E. australis* (GeneArt, Thermo Fisher Scientific) was cloned into a modified pRSETA vector (Invitrogen, Thermo Fisher Scientific) using conventional restriction digestion and ligation protocols. Recombinant CTD was overexpressed in *E. coli* C41 (DE3) bacterial cells as His-tagged, C-terminal fusion protein. The lipoyl domain of a pyruvate dehydrogenase multienzyme complex was used as a fusion protein. Single-point mutants were generated using the QuikChange mutagenesis protocol (Stratagene). Overexpressed CTD and mutants thereof were isolated from clarified cell lysate by affinity chromatography using Ni-Sepharose 6 Fast-Flow resin (GE Healthcare) followed by proteolytic cleavage of the His$_6$-tagged fusion protein using thrombin from bovine plasma (Sigma-Aldrich). The dialyzed thrombin digest was again applied to Ni-Sepharose chromatography to remove the His-tagged fusion protein. The isolated CTD was purified to homogeneity using size exclusion chromatography (SEC) on a Superdex 75 column (GE Healthcare) in 200 mM ammonium bicarbonate buffer. Pooled protein fractions were lyophilized. Purity of protein samples was confirmed by sodium dodecyl sulfate polyacrylamide gel electrophoresis. Wild-type CTD and mutant C82A-S14C were modified using thiol-reactive maleimide derivatives of the fluorophores Alexa-Fluor-488, Alexa-Fluor-594 (both Thermo Fisher Scientific), and AttoOxa11 (AttoTec). Labeling was carried out using a fivefold molar excess of fluorophore incubated for 3 h at 298 K in 50 mM phosphate buffer, pH 7.5, with the ionic strength adjusted to 200 mM using potassium chloride, and a tenfold molar excess of tris(2-carboxyethyl)phosphine (TCEP) used to reduce the native disulfide or to prevent thiol oxidation of mutant C82A–S14C.

Labeled protein was isolated from nonreacted dye using SEC (Sephadex G-25 resin, GE Healthcare).

**Far-UV CD spectroscopy**. Far-UV CD spectroscopy was performed using a J-815 spectropolarimeter (Jasco). For the recording of steady-state spectra and for equilibrium denaturation experiments a 1 mm pathlength cuvette (Hellma) was used. The typical protein concentration was 10 or 15 μM CTD, except for concentration-dependent experiments. In equilibrium denaturation experiments the CD signal was recorded at 222 nm probing α-helix secondary structure. Denaturation experiments were performed by manual titration between zero and 10 M urea in 50 mM phosphate buffer, pH 7.0, with the ionic strength adjusted to 200 mM using potassium chloride, or in 20 mM 2-(N-morpholino)ethanesulfonic acid buffer, pH 5.7, with the ionic strength adjusted to 200 mM using potassium chloride, and at 298 K. Thermal denaturation was carried out using the same buffers and applying a temperature ramp at a rate of 1 K min$^{-1}$. Samples for experiments carried out under reducing conditions contained 1 mM dithioerythritol (DTE) in pH 7.0 buffered solutions, or 1 mM TCEP in pH 5.7 buffered solutions.

Stopped-flow far-UV CD spectroscopy was carried out on a SFM-2000 stopped-flow machine (BioLogic Instruments) coupled to a J-815 spectropolarimeter (Jasco) at 298 K. Stopped-flow CD transients were recorded at 222 nm. In initial refolding experiments aimed at probing the transition D to N$_2$, fully denatured CTD in 8 M guanidinium chloride containing 10 mM DTE was mixed into 50 mM phosphate buffer, pH 7.0, with the ionic strength adjusted to 200 mM using potassium chloride, using a mixing ratio of 1:20. For unfolding/refolding experiments probing the transition N$_2$ to I$_2$ and vice versa, 0.5 mM CTD samples were rapidly mixed into 50 mM phosphate buffer, pH 7.0, with the ionic strength adjusted to 200 mM using potassium chloride, containing various concentrations of urea. The applied mixing ratio was 1:10. Typically ten shots were averaged to obtain transients of sufficient signal-to-noise for analysis.

**Absorption and fluorescence spectroscopy**. Absorption and fluorescence emission spectra were recorded on a V-650 spectrophotometer (Jasco) and on a FP-6500 spectrofluorometer (Jasco) at 298 K using a 1-cm pathlength quartz glass cuvette (Hellma). FRET emission spectra were recorded from mixtures of 100 nM Alexa-Fluor-488-labeled (donor) and 500 nM Alexa-Fluor-594-labeled (acceptor) mutant C82A-S14C, which were initially denatured in 8 M urea and refolded in 50 mM phosphate buffer, pH 7.0, with the ionic strength adjusted to 200 mM using potassium chloride, to form hetero-dimers. Alexa-Fluor-488 fluorescence was excited at 460 nm. Chemical equilibrium denaturation was performed by manual titration between zero and 10 M urea using the same buffer. The equilibrium denaturation curve was obtained by excitation of Alexa-Fluor-488 at 460 nm and recording the fluorescence emission of Alexa-Fluo-594 at 616 nm, i.e., the wavelength of maximal emission intensity. AttoOxa11-modified CTD was excited at 610 nm and emission spectra were recorded in the buffer described above. In chasing experiments, 60, 120, and 180 nM CTD-AttoOxa11 were chased with 20 μM chemically reduced wild-type CTD by manual mixing in a 1-cm pathlength quartz glass cuvette at 298 K. Fluorescence intensity time traces were recorded at the wavelength of maximal emission intensity (at 677 nm) using a FP-6500 spectrofluorometer (Jasco).

Stopped-flow fluorescence spectroscopy of CTD dimerization was measured on a SFM-2000 machine (BioLogic Instruments) equipped with a 639-nm diode as excitation source and a photomultiplier tube as fluorescence detector. CTD-AttoOxa11 samples were prepared in 50 mM phosphate buffer, pH 7.0, with the ionic strength adjusted to 200 mM using potassium chloride, containing 8 M urea to form denatured monomers, and mixed into the same buffer without urea to re-assemble CTD-AttoOxa11 dimers. The mixing ratio on the stopped-flow machine was 1:10. All measurements were performed at 298 K.

**Data analysis**. Chemical equilibrium denaturation data of wild-type CTD recorded under non-reducing conditions using far-UV CD spectroscopy and of Alexa-Fluor-488/Alexa-Fluor-594-modified mutant C82A-S14C using FRET spectroscopy, i.e., data sets that showed one transition between native dimer N$_2$ and dimeric intermediate I$_2$, were fitted using the thermodynamic model for a two-state equilibrium. In this model, the spectroscopic signal S is expressed as a function of denaturant concentration ([urea])[50]:

$$S([\text{urea}]) = \frac{\alpha_{N2} + \beta_{N2} \cdot [\text{urea}] + (\alpha_{I2} + \beta_{I2} \cdot [\text{urea}]) \cdot \exp(-\Delta G_{N2-I2}([\text{urea}])/RT)}{1 + \exp(-\Delta G_{N2-I2}([\text{urea}])/RT)}, \quad (1)$$

where $\alpha_{N2}$, $\beta_{N2}$, $\alpha_{I2}$, and $\beta_{I2}$ are the linearly sloping baselines of native dimer and dimeric intermediate, R is the gas constant, T the temperature ($T = 298$ K), and $\Delta G_{N2-I2}$ is the difference in free energy between N$_2$ and I$_2$. In protein chemical denaturation experiments the linear-free energy relationship applies[23]:

$$\Delta G_{N2-I2}([\text{urea}]) = \Delta G_{N2-I2} - m_{N2-I2} \cdot [\text{urea}], \quad (2)$$

where $m_{N2-I2}$ is the equilibrium m-value that describes the sensitivity of the folding equilibrium to denaturant. Experimental errors of $\Delta G_{N2-I2}$ were determined from

propagated errors of fitted values of $m_{N2-I2}$ and mid-point concentrations of urea ([urea]$_{50\%}$).

Chemical equilibrium denaturation data of wild-type CTD recorded under reducing conditions and those of mutants C82A and C82A–S14C were fitted using a thermodynamic model for a three-state equilibrium between a native dimer N$_2$, a dimeric intermediate I$_2$, and denatured monomers D, detailed elsewhere[21]. The spectroscopic signal is expressed as the sum of the signals of N$_2$ ($S_{N2}$), I$_2$ ($S_{I2}$), and D ($S_D$):

$$S = S_{N2}\left(\frac{2 \cdot P_t \cdot F_D^2}{K_1 \cdot K_2}\right) + S_{I2}\left(\frac{2 \cdot P_t \cdot F_D^2}{K_2}\right) + S_D(F_D). \quad (3)$$

The fraction of denatured monomer, $F_D$, is expressed as[21]:

$$F_D = \frac{-K_1 \cdot K_2 + \sqrt{(K_1 \cdot K_2)^2 + 8 \cdot (1 + K_1)(K_1 \cdot K_2) \cdot P_t}}{4 \cdot P_t(1 + K_1)}. \quad (4)$$

The equilibrium constant of the first transition (N$_2$ to I$_2$), $K_1$, is expressed as:

$$K_1 = \exp\left(\frac{m_{N2-I2} \cdot ([\text{urea}] - [\text{urea}]_{50\%})}{RT}\right). \quad (5)$$

The equilibrium constant of the second transition (I$_2$ to D), $K_2$, is expressed as:

$$K_2 = \exp\left(\frac{-\Delta G_{I2-D} + m_{I2-D} \cdot [\text{urea}]}{RT}\right). \quad (6)$$

In Eqs. (3) and (4), $P_t$ is the total protein concentration in terms of monomer. In Eq. (6) $\Delta G_{I2-D}$ and $m_{I2-D}$ are the free energy and equilibrium m-value of the second transition I$_2$ to D.

Thermodynamic parameters derived from far-UV CD chemical denaturation data of wild-type CTD measured under reducing conditions are mean values of four measurements recorded at different protein concentrations (5, 10, 15, and 20 μM). Errors are standard deviations of the four measurements. Other errors are standard errors from regression analysis and propagated standard errors.

The kinetic transients of the folding/unfolding transitions N$_2$ to I$_2$ measured using stopped-flow far-UV CD spectroscopy were fitted to a mono-exponential decay function:

$$S(t) = a \cdot \exp(-k_{obs} \cdot t) + b. \quad (7)$$

$S(t)$ is the fluorescence signal as function of time, a is the amplitude, $k_{obs}$ is the observed rate constant, and b is the baseline signal. $k_{obs}$ contains the sum of microscopic rate constants of folding and unfolding ($k_f$ and $k_u$). $k_{obs}$ as a function of denaturant concentration was analyzed by fitting the data set to the chevron model for a barrier-limited two-state transition containing the linear-free-energy relationship[27]:

$$\log k_{obs}([\text{urea}]) = \log[k_f \exp(-m_f[\text{urea}]/RT) + k_u \exp(m_u[\text{urea}]/RT)], \quad (8)$$

$m_f$ and $m_u$ are the kinetic m-values of barrier-limited folding and unfolding. $k_f$ and $k_u$ are the microscopic rate constants of folding and unfolding, respectively, under standard solvent conditions and in the absence of denaturant.

Kinetics of CTD dimerization were obtained from fitting a reaction model of protein dimerization to transients[51]:

$$2D \overset{k_{ass}}{\leftrightarrow} I_2, \quad (9)$$

$$\frac{d[I_2]}{dt} = k_{ass}[D]^2, \quad (10)$$

where $k_{ass}$ is the microscopic, bimolecular rate constant of dimerization. The differential equation can be solved to give[51]:

$$S(t) = S_{t=0} + S\frac{(k_{obs}t)}{(1 + k_{obs}t)}, \quad (11)$$

where $S(t)$ is the time-dependent signal, $S_{t=0}$ is the signal at time $t = 0$, S is the signal change, and $k_{obs}$ is the observed rate constant. The observed rate constant is related to the microscopic rate constant $k_{ass}$ (ref. [51]):

$$k_{obs} = c_D \cdot k_{ass}, \quad (12)$$

where $c_D$ is the protein concentration in terms of denatured monomer.

Rate constants of dissociation were obtained from fitting kinetic transients of chasing experiments to a bi-exponential function:

$$S(t) = a_1 \exp(-t/t_1) + a_2 \exp(-t/t_2) + b. \qquad (13)$$

$a_1$ ($t_1$) and $a_2$ ($t_2$) are the observed amplitudes (time constants) of the first and of the second decay, and $b$ is the observed baseline signal.

Errors of values derived from kinetic experiments are standard errors from regression analysis and propagated standard errors.

## Data availability

The data that support the findings of this study are available from the corresponding author upon reasonable request.

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

## Acknowledgments

We thank Sören Doose for programming the software tool used to create theoretical three-state denaturation data. The authors are grateful to the U.S. Army Research Office for financial support (Grant number W911NF-17-1-0336).

## Author contributions

C.R. synthesized and modified protein material, performed chemical and thermal denaturation experiments, performed far-UV CD and stopped-flow fluorescence experiments and analyzed data. J.C.H. synthesized protein material, performed far-UV CD stopped-flow experiments and analyzed data. J.P.B. synthesized protein material, performed chemical denaturation experiments, far-UV CD spectroscopy, and analyzed data. H.N. designed experiments, analyzed data, and wrote the paper.

## Additional information

**Competing interests:** The authors declare no competing interests.

