## [Peer Review File · Nature Communications]

Reviewers' comments:

Reviewer #1 (Remarks to the Author):

Spider silk, as often noted, has a tensile strength which exceeds that of steel. The material is produced on demand within a fraction of a second, starting from unstructured, soluble monomers stored in the ampulla, situated in the spider's abdomen. Like blood clotting in warm-blooded animals, prevention of premature fiber formation in the ampulla is of critical physiological importance. Consequently, the mechanism of silk formation is of both biophysical and bioengineering interest.

The biophysics of spider silk assembly has been studied extensively, as noted in this manuscript. The fibroin monomer is flanked by N- and C-terminal domains, each of which comprises a folded, five-helix bundle. In dragline silk, this monomer dimerizes, and the N- and C-terminal domains are involved in the mechanism of dimerization. Dimerization of the C-terminal domain is of central importance in this self-assembly process. Currently, the C-terminal domain is thought to fold in a two-state manner. This manuscript challenges the current model, presenting both thermodynamic and kinetic evidence in favor of a three-state mechanism of dimerization. Assembly thermodynamics was based on reversible denaturation studies under suitable conditions. Assembly kinetics was deduced from both FRET and contact-induced fluorescence quenching and from circular dichroism spectroscopy.

The analysis detected a C-terminal domain folding intermediate, and a two-step assembly model was formulated, consistent with the wealth of kinetic data. Reassuringly, a slow-step in the model, which might seem incompatible with the known sub-second assembly of silk, provides an inherent lability that might explain silk elasticity.

Historically, protein folding studies have emphasized two-state folders that lack populated intermediates. More complicated folding mechanisms often had to await the advent of contemporary spectroscopic methods, of which the corresponding author is an expert.

Here are a couple of trivial issues, to show I read the paper carefully:

pg. 5 This is not a complete sentence. "Although the disulfide-containing dimer resisted full unfolding in chemical denaturation experiments, heat unfolded residual structure at a mid-transition temperature of ~80 °C (Supplementary Figure 2)."

pg. 7 Serves, not 'serving'. "... shows a characteristic $1/r^6$ distance-dependence and serving as a ..."

The paper is well written and a pleasure to read,

Reviewer #2 (Remarks to the Author):

The manuscript entitled "Two-step self-assembly of a spider silk molecular clamp" proposes the elucidation of the folding process of the C-terminal dimeric domain of a spidroin. The scientific approach is clearly presented.

The NMR structure of the protein domain studied has been published previously. The authors present an elegant study using a combination of different methods: equilibrium and time-scale folding experiments with circular dichroism, fluorescence spectroscopy and FRET. The results obtained are convincing and show that the first step of the folding process is the dimerization of the protein with the folding of the helices 4 and 5, the helices 4 being at the interface between the

two monomers. Then, in a second step the helices 1 to 3 fold, and lock the structure. Hence, the formation of a dimeric intermediate could be observed.

Major comments:

The three-state mechanism of self-assembly elucidated in this study has not been discovered before as most of previous studies (ref 6 and 14-16 cited by the authors) focused on the effect of pH and salts, parameters known to be important for silk self-assembly process in vivo. In the study presented here, the authors use urea and reductant as denaturants to detect this three-state mechanism of self-assembly. These denaturants are useful tools to analyze the folding process of a protein in vitro but these conditions have not been detected in the gland or spinning duct of the spider. The link with the in vivo process is missing and it could be interesting to prove the importance of this self-assembly mechanism for the silk formation.

The authors show that the folding and self-assembly process of the protein resembles the mechanism of knotted proteins, a family of proteins which is well known.

Based on the results obtained, the authors suggest that the helices 1-3 of C-terminal domain could play a role in the extensibility of spider silk (page 17, line 19). To prove that, it would be interesting to perform a study on recombinant Ma spidroins and create mutants with mutations in this helices 1-3 region (for example, truncated mutants or with different folding kinetics). And then the authors could test the impact of these mutations on the extensibility of the silk fibers made of these mutated proteins.

Although the experiments performed are rigorous and the results obtained convincing, the folding and self-assembly process elucidated in this study is similar to that of other complex known proteins and the demonstration of the importance of these results for the silk formation in vivo is missing.

Minor comments:

-introduction, p3, line 10: not all spidroins sequences are composed of poly-alanine and glycine-rich stretches (see for example: Babb et al – Nat Genet. 2017, 49, 6, 895-903). This sentence is slightly limiting. The authors should precise the type of spidroin they want to refer.

-introduction, p3, line 12: in the literature on spider silk, a lot of papers mention the highly conserved feature of the N and C-terminal domains. However, a recent paper on a big analysis of 145 spidroin terminal domains tends to show that these sequences are not highly conserved (see Collin et al Int J Biol Macromol. 2018 113, 829-840). I suggest that the authors correct their manuscript according to this recent publication.

-results, p7, line 3: the reference 16 is not properly indicated.

-Figure 1: on figure 1c, the surimposition of the two spectra of the protein at pH 7 and pH 5.7 is obvious. It is possible to see that the CD signal at 222 nm is about -40 mdeg; However, in the denaturation experiment, figure 1d, the CD signal at 222 nm and at 0M urea is about -27 mdeg showing that the experiment has been performed in different concentrations conditions or that the protein was previously denatured. Could the authors explain that difference?

-The same remark could be made for the figures 2b and 2c. In figure 2b, the intensity of the fluorescence peak at 615 nm reaches about 200 a.u. However in figure 2c, in the native condition at 0M of urea, the fluorescence intensity at 615 nm is about 85 a.u.

-In Figure 2: the authors should precise the pdb code of the protein used to create the cartoon representation.

Reviewer #3 (Remarks to the Author):

The manuscript describes a thorough investigation of the unfolding and folding mechanism of the C-terminal domain of spidroins. The text is overall well written, the data are solid and presented in a clear way.

Major comments:

1. Think through when it is appropriate to use the words; self-assembly, assembly, dimerization and folding. In my opinion the data presented herein gives valuable insights about folding and dimerization of CTD, but that the words self-assembly and assembly are sometimes not used in an appropriate way.
2. It would be helpful for the reader if an illustration of the hypothesized folding steps could be included, for example in figure 1.
3. Complementary studies using either far UV CD or SAXS would strengthen the hypothesis of an extended structure in the partly unfolded state.
4. Have you verified that the labeling of CTD with fluorophore did not alter the structures? CD? Molecular modelling?

Specific comments:

1. If the expression "molecular clamp" should be used in the title it must be better clarified in the manuscript.
2. Page 3, avoid the word fibroin (from silkworm) when talking about spidroins (from spiders).
3. Page 3, Is it really correct to describe silk formation performed by the spider as "synthesis"?
4. Page 4, Re-think if the expression "highly expanded state" is clear to the reader.
5. Page 5, Rephrase the sentence with "heat unfolded residual structure"
6. Page 6, Rephrase the sentence with "we stepped back"
7. Page 6, "reducing solution conditions" Remove solution.
8. Page 7, Relate the observed effect of pH with the salt bridges present in the structure.
9. Page 12, Rephrase the sentence with "disulphide is in fully"
10. Page 14, Are the antibodies used in ref 31 fold-dependent?
11. Page 14, Include ref for the previous studies stated to be in agreement.
12. Page 18, Clarified that the proteins where expressed recombinantly and not synthesized.
13. Fig. 1c, Is this data from non-reducing conditions?
14. Fig. 4a, Mark helices from both monomers.
15. Some references have not been included properly using the reference program.

Response to the referee comments on NCOMMS-18-24971

We thank all three referees for their time and effort reviewing our manuscript and for their valuable and constructive comments. In the following, we address their comments (cited in *italics*) point by point and highlight changes made in the revised manuscript.

Reviewer 1:

Comments:

Spider silk, as often noted, has a tensile strength which exceeds that of steel. The material is produced on demand within a fraction of a second, starting from unstructured, soluble monomers stored in the ampulla, situated in the spider's abdomen. Like blood clotting in warm-blooded animals, prevention of premature fiber formation in the ampulla is of critical physiological importance. Consequently, the mechanism of silk formation is of both biophysical and bioengineering interest.

The biophysics of spider silk assembly has been studied extensively, as noted in this manuscript. The fibroin monomer is flanked by N- and C-terminal domains, each of which comprises a folded, five-helix bundle. In dragline silk, this monomer dimerizes, and the N- and C-terminal domains are involved in the mechanism of dimerization. Dimerization of the C-terminal domain is of central importance in this self-assembly process. Currently, the C-terminal domain is thought to fold in a two-state manner. This manuscript challenges the current model, presenting both thermodynamic and kinetic evidence in favor of a three-state mechanism of dimerization. Assembly thermodynamics was based on reversible denaturation studies under suitable conditions. Assembly kinetics was deduced from both FRET and contact-induced fluorescence quenching and from circular dichroism spectroscopy.

The analysis detected a C-terminal domain folding intermediate, and a two-step assembly model was formulated, consistent with the wealth of kinetic data. Reassuringly, a slow-step in the model, which might seem incompatible with the known sub-second assembly of silk, provides an inherent lability that might explain silk elasticity.

Historically, protein folding studies have emphasized two-state folders that lack populated intermediates. More complicated folding mechanisms often had to await the advent of contemporary spectroscopic methods, of which the corresponding author is an expert.

Here are a couple of trivial issues, to show I read the paper carefully:

pg. 5 This is not a complete sentence. "Although the disulfide-containing dimer resisted full unfolding in chemical denaturation experiments, heat unfolded residual structure at a mid-transition temperature of ~80 °C (Supplementary Figure 2)."

pg. 7 Serves, not 'serving'. " ... shows a characteristic 1/r⁶ distance-dependence and serving as a ..."

The paper is well written and a pleasure to read.

Response:

We thank the reviewer for this positive assessment of our work.

In response to the minor comments, we changed the sentence on page 6 of the revised manuscript to “Although the dimeric CTD containing a disulfide bond resisted full unfolding in chemical denaturation experiments, thermal denaturation lead to complete unfolding of residual structure at a mid-transition temperature of ~80 °C”, and changed “serving” to “serves” in the sentence on page 8 of the revised manuscript, as requested by the reviewer.

Reviewer 2:

Comment:

The manuscript entitled “Two-step self-assembly of a spider silk molecular clamp” proposes the elucidation of the folding process of the C-terminal dimeric domain of a spidroin. The scientific approach is clearly presented.

The NMR structure of the protein domain studied has been published previously. The authors present an elegant study using a combination of different methods: equilibrium and time-scale folding experiments with circular dichroism, fluorescence spectroscopy and FRET. The results obtained are convincing and show that the first step of the folding process is the dimerization of the protein with the folding of the helices 4 and 5, the helices 4 being at the interface between the two monomers. Then, in a second step the helices 1 to 3 fold, and lock the structure. Hence, the formation of a dimeric intermediate could be observed.

Response:

We thank the reviewer for this positive assessment of our work.

Comment:

Major comments:

The three-state mechanism of self-assembly elucidated in this study has not been discovered before as most of previous studies (ref 6 and 14-16 cited by the authors) focused on the effect of pH and salts, parameters known to be important for silk self-assembly process in vivo. In the study presented here, the authors use urea and reductant as denaturants to detect this three-state mechanism of self-assembly. These denaturants are useful tools to analyze the folding process of a protein in vitro but these conditions have not been detected in the gland or spinning duct of the spider. The link with the in vivo process is missing and it could be interesting to prove the importance of this self-assembly mechanism for the silk formation.

The authors show that the folding and self-assembly process of the protein resembles the mechanism of knotted proteins, a family of proteins which is well known. Based on the results obtained, the authors suggest that the helices 1-3 of C-terminal domain could play a role in the extensibility of spider silk (page 17, line 19). To prove that, it would be interesting to perform a study on recombinant Ma spidroins and create mutants with mutations in this helices 1-3 region (for example, truncated mutants or with different folding kinetics). And then the authors could test the impact of these mutations on the extensibility of the silk fibers made of these mutated proteins.

Although the experiments performed are rigorous and the results obtained convincing, the folding and self-assembly process elucidated in this study is similar to that of other complex known

proteins and the demonstration of the importance of these results for the silk formation in vivo is missing.

Response:

We agree with the reviewer that chemical denaturation experiments apply non-physiological conditions, which also holds true for thermal denaturation where proteins are unfolded at non-physiological temperature. But analysis of chemical denaturation data commonly involves application of the linear free-energy relationship (LFER, i.e. the linear dependence of free energy on denaturant concentration) and this was the case in our study. Thermodynamic and kinetic quantities derived from the analysis of chemical denaturation data involving the LFER are consequently values extrapolated to physiological solution conditions. In our case, this was 50 mM phosphate pH 7.0 containing 200 mM salt, conditions that resemble the ones in the ampulla of a spinning gland where the CTD folds and dimerizes. The quantities thus represent the physiological relevant values.

We performed extensive mutagenesis experiments on the CTD to also try to do exactly what the reviewer suggested, i.e. to make N-terminal helices more labile. The tested single-point mutants involved R42A and E91Q, which were designed to remove the salt bridge that stabilizes helix 2. Mutant L34A was designed to remove a hydrophobic interaction that stabilizes helix 1. Unfortunately, expression and isolation of many single-point mutants failed because of protein aggregation. We found that the *E. australis* MaSp1 CTD was very sensitive to mutation. Mutants described in the manuscript were the cases where site-directed mutagenesis yielded stably folded and soluble protein material.

We agree with the reviewer that it would be desirable to test our results in in-vivo experiments or in material. However, the direct observation of structural transitions of CTDs in the context of an intact spidroin is complicated because of background signal arising from the NTD that has the same secondary structure. Further, some areas in the repetitive segments that are also thought to form helix. Laboratory synthesis of spider silk material from engineered spidroins is currently beyond our laboratory expertise. But we anticipate that our study will stimulate experiments by material scientists designing engineered spidroins containing mutant or truncated CTDs in order to elucidate effects in the material. Of note, Gnesa et al. (Biomacromolecules 2012, 13, 304-312; ref. 46 in the revised manuscript) already reported that synthetic spider silk fibers lacking the CTD showed reduced extensibility, which supports our interpretation. We discussed this on page 16 of the revised manuscript.

Comment:

Minor comments:

-introduction, p3, line 10: not all spidroins sequences are composed of poly-alanine and glycine-rich stretches (see for example: Babb et al – Nat Genet. 2017, 49, 6, 895-903). This sentence is slightly limiting. The authors should precise the type of spidroin they want to refer.

Response:

We now specified at the beginning of the paragraph on page 3 of the revised manuscript that we refer to spidroins from the major ampullate (Ma) gland, with reference to ref. 4: “Spidroins from the major ampullate (Ma) gland form dragline silk, which represents the toughest fiber used as a

lifeline or to build a web frame. Ma silk is a current focus of biomimetic material sciences. The central segments of Ma spidroins consist of repetitive peptide motifs of simple amino acid composition involving alanine-, glycine-, and proline-rich stretches, which are unstructured under storage conditions and form mainly β -sheet structure in fibers (ref. 4).”

Comment:

-introduction, p3, line 12: in the literature on spider silk, a lot of papers mention the highly conserved feature of the N and C-terminal domains. However, a recent paper on a big analysis of 145 spidroin terminal domains tends to show that these sequences are not highly conserved (see Collin et al Int J Biol Macromol. 2018 113, 829-840). I suggest that the authors correct their manuscript according to this recent publication.

Response:

We added the sentence “A recent genomic study shows that sequences of CTDs are slightly more diverse compared to those of NTDs. In particular, sequence of CTDs from the pyriform and aggregate glands differ from those of other glands, which may be explained by the fact pyriform and aggregate silk has adhesive rather than fiber-forming function” to page 3 of the revised manuscript, referring to Collin et al. (Ref. 5 in the revised manuscript) as requested by the reviewer.

Comment:

-results, p7, line 3: the reference 16 is not properly indicated.

Response:

We reformatted the mentioned reference.

Comment:

-Figure 1: on figure 1c, the surimposition of the two spectra of the protein at pH 7 and pH 5.7 is obvious. It is possible to see that the CD signal at 222 nm is about -40 mdeg; However, in the denaturation experiment, figure 1d, the CD signal at 222 nm and at 0M urea is about -27 mdeg showing that the experiment has been performed in different concentrations conditions or that the protein was previously denatured. Could the authors explain that difference?

Response:

Denaturation data in Figure 1d were recorded at 10 μ M protein concentration, which resulted in a lower amplitude of native-state ellipticity. We now specified that data in panel 1d were recorded at 10 μ M protein concentration in the legend of Figure 1 of the revised manuscript.

Comment:

-The same remark could be made for the figures 2b and 2c. In figure 2b, the intensity of the fluorescence peak at 615 nm reaches about 200 a.u. However in figure 2c, in the native condition at 0M of urea, the fluorescence intensity at 615 nm is about 85 a.u.

Response:

The absolute fluorescence intensities reported in Figure 2 depend on protein concentration but also on instrument settings (i.e. the variable slit widths of excitation and detection light paths). Instruments settings can differ in recordings carried out on different days and using different experimental design. The titles of the y-axes in Figures 2b and 2c therefore specify the measured fluorescence intensities as arbitrary units (a.u.). The important information in Figures 2b and 2c is the relative peak intensities of donor and acceptor fluorescence and the change of acceptor fluorescence with increasing concentration of denaturant. This information is independent on the absolute measured intensities.

Comment:

-In Figure 2: the authors should precise the pdb code of the protein used to create the cartoon representation.

Response:

We now specified in the legend of Figure 2 of the revised manuscript the pdb code (2KHM) of the structure used as a template for the homology model. Template structure and homology model are also specified in the main text on page 8 of the revised manuscript.

Reviewer 3:

Comment:

The manuscript describes a thorough investigation of the unfolding and folding mechanism of the C-terminal domain of spidroins. The text is overall well written, the data are solid and presented in a clear way.

Response:

We thank the reviewer for this positive assessment of our work.

Comment:

Major comments:

1. Think through when it is appropriate to use the words; self-assembly, assembly, dimerization and folding. In my opinion the data presented herein gives valuable insights about folding and dimerization of CTD, but that the words self-assembly and assembly are sometimes not used in an appropriate way.

Response:

We agree with the reviewer that the words “self-assembly” and “assembly” were not always used correctly. The word “self-assembly” describes “folding and dimerization”. We often wrote “folding and self-assembly”, which was incorrect, and now replaced the phrase by “folding and dimerization” where appropriate. We went through the entire text and changed the wording accordingly at positions highlighted in the revised manuscript.

Comment:

2. It would be helpful for the reader if an illustration of the hypothesized folding steps could be included, for example in figure 1.

Response:

We agree with the reviewer that an additional illustration of the folding steps is a good idea. Since the full picture of folding and dimerization requires data from the entire results section we revised Figure 5 in the discussion section and included a cartoon representation of the individual steps in new Figure 5a of the revised manuscript. As a result of this revision, original Figure 5b is now included in the cartoon in new Figure 5a. Original Figure 5c is now Figure 5b. We amended the legend of Figure 5 and the numbering of Figures in the discussion section accordingly.

Comment:

3. Complementary studies using either far UV CD or SAXS would strengthen the hypothesis of an extended structure in the partly unfolded state.

Response:

We agree that additional SAXS experiments would be helpful to further strengthen the hypothesis of an extended structure. However, the SAXS methodology is beyond our laboratory expertise and instrumentation. We characterized the intermediate using far-UV CD spectroscopy, which showed a helix-to-coil transition from the native dimer to the intermediate dimer, and by FRET spectroscopy providing distance estimates of chain ends in the dimeric intermediate. We further estimated chain dimensions using polymer theory. We believe that the collated results provide sufficient evidence for the presence of an extended structure.

Comment:

4. Have you verified that the labeling of CTD with fluorophore did not alter the structures? CD? Molecular modelling?

Response:

We have experimental evidence that fluorescence modification did not perturb structure or energetics. In case of FRET labels, thermodynamic quantities derived from denaturation data measured using FRET ($m = 1.0 \pm 0.1 \text{ kcal mol}^{-1} \text{ M}^{-1}$, $[\text{urea}]_{50\%} = 2.5 \pm 0.1 \text{ M}$, $\Delta G_{\text{N2-I2}} = 2.5 \pm 0.3 \text{ kcal mol}^{-1}$) matched values measured using non-modified, mutant protein material by far-UV CD spectroscopy ($m = 1.0 \pm 0.1 \text{ kcal mol}^{-1} \text{ M}^{-1}$, $[\text{urea}]_{50\%} = 2.5 \pm 0.1 \text{ M}$, $\Delta G_{\text{N2-I2}} = 2.5 \pm 0.3 \text{ kcal mol}^{-1}$). These values were in turn within error with quantities of non-reduced wild-type protein measured using far-UV CD spectroscopy ($m = 1.2 \pm 0.2 \text{ kcal mol}^{-1} \text{ M}^{-1}$, $[\text{urea}]_{50\%} = 2.2 \pm 0.1 \text{ M}$, $\Delta G_{\text{N2-I2}} = 2.6 \pm 0.5 \text{ kcal mol}^{-1}$). We added the sentence “The good agreement of thermodynamic quantities derived from modified and non-modified material showed that fluorescence modification did not perturb folding” to page 9 of the revised manuscript. In case of contact-induced self-quenching of AttoOxa11-modified wild-type protein, the spectroscopic signature of the fluorophore dimer seen in Figure 4b showed that the modified cysteine was contact-distance similar as in the structure. A lack of perturbation was also evident from comparing the free energy of dimerization calculated from kinetics measured using the fluorescently modified protein ($\Delta G = -RT \ln(k_{\text{ass}}/k_{\text{diss}}) = 13.4 \pm 0.1 \text{ kcal mol}^{-1}$) with the free energy obtained from equilibrium chemical denaturation of wild-type protein measured using far-UV CD spectroscopy ($\Delta G_{\text{I2-D}} = 14 \pm 1 \text{ kcal mol}^{-1}$). The values were

within error. We discussed this on page 13 at the end of the results section of the revised manuscript.

Comment:

Specific comments:

1. *If the expression “molecular clamp” should be used in the title it must be better clarified in the manuscript.*

Response:

We now specified the term “molecular clamp” by adding the sentence “The topology of the fold resembles a molecular clamp” to the introduction section on page 4 of the revised manuscript.

Comment:

2. *Page 3, avoid the word fibroin (from silkworm) when talking about spidroins (from spiders).*

Response:

We deleted “fibroin” and changed the sentence to “...the constituting protein building blocks, called spidroins, ...”.

Comment:

3. *Page 3, Is it really correct to describe silk formation performed by the spider as “synthesis”?*

Response:

We replaced “synthesized” by “formed”.

Comment:

4. *Page 4, Re-think if the expression “highly expanded state” is clear to the reader.*

Response:

We replaced “state” with “structure”.

Comment:

5. *Page 5, Rephrase the sentence with “heat unfolded residual structure”*

Response:

We changed the sentence to “...thermal denaturation lead to complete unfolding of residual structure at a mid-transition temperature of ~80 °C”.

Comment:

6. *Page 6, Rephrase the sentence with “we stepped back”*

Response:

We replaced “we stepped back” by “we omitted”.

Comment:

7. *Page 6, “reducing solution conditions” Remove solution.*

Response:

We deleted “solution”.

Comment:

8. Page 7, *Relate the observed effect of pH with the salt bridges present in the structure.*

Response:

To rationalize the reduced stability of the first unfolding transition at pH 5.7, we added the sentence “The observation may be explained by weakening of the salt bridge formed by residues Arg42 and Glu91 at low pH through protonation, which stabilizes N-terminal helix 2 of the fold”.

Comment:

9. Page 12, *Rephrase the sentence with “disulphide is in fully”.*

Response:

We deleted “fully” on page 12 of the manuscript.

Comment:

10. Page 14, *Are the antibodies used in ref 31 fold-dependent?*

Response:

Ref. 31 does not specify whether the antibodies are fold-dependent or not. Given this uncertainty, we deleted the sentence referring to ref. 31 on page 14 of the revised manuscript.

Comment:

11. Page 14, *Include ref for the previous studies stated to be in agreement.*

Response:

We included refs. 15 and 17 on page 14 in the revised manuscript to specify previous studies.

Comment:

12. Page 18, *Clarified that the proteins where expressed recombinantly and not synthesized.*

Response:

We changed “protein synthesis” to “protein expression” in the title of the first paragraph of the methods section.

Comment:

13. Fig. 1c, *Is this data from non-reducing conditions?*

Response:

These data were measured under non-reducing conditions. We specified this in the revised manuscript in the legend of Figure 1.

Comment:

14. Fig. 4a, *Mark helices from both monomers.*

Response:

We now marked all helices in Figure 4a of the revised manuscript.

Comment:

15. Some references have not been included properly using the reference program.

Response:

We revised the list of references and hopefully deleted all mistakes.

REVIEWERS' COMMENTS:

Reviewer #2 (Remarks to the Author):

I am fully satisfied with the answers of the authors.

Reviewer #3 (Remarks to the Author):

The revised version of the manuscript is now suitable for publication.